# Alteration of Genomic Imprinting after Assisted Reproductive Technologies and Long-Term Health

**DOI:** 10.3390/life11080728

**Published:** 2021-07-22

**Authors:** Eguzkine Ochoa

**Affiliations:** Department of Medical Genetics, University of Cambridge and NIHR Cambridge Biomedical Research Centre, Cambridge CB2 0QQ, UK; eo309@medschl.cam.ac.uk; Tel.: +44-1233-746714

**Keywords:** genomic imprinting, assisted reproductive technology, DNA methylation, epigenetic reprogramming, long-term health

## Abstract

Assisted reproductive technologies (ART) are the treatment of choice for some infertile couples and even though these procedures are generally considered safe, children conceived by ART have shown higher reported risks of some perinatal and postnatal complications such as low birth weight, preterm birth, and childhood cancer. In addition, the frequency of some congenital imprinting disorders, like Beckwith–Wiedemann Syndrome and Silver–Russell Syndrome, is higher than expected in the general population after ART. Experimental evidence from animal studies suggests that ART can induce stress in the embryo and influence gene expression and DNA methylation. Human epigenome studies have generally revealed an enrichment of alterations in imprinted regions in children conceived by ART, but no global methylation alterations. ART procedures occur simultaneously with the establishment and maintenance of imprinting during embryonic development, so this may underlie the apparent sensitivity of imprinted regions to ART. The impact in adulthood of imprinting alterations that occurred during early embryonic development is still unclear, but some experimental evidence in mice showed higher risk to obesity and cardiovascular disease after the restriction of some imprinted genes in early embryonic development. This supports the hypothesis that imprinting alterations in early development might induce epigenetic programming of metabolism and affect long-term health. Given the growing use of ART, it is important to determine the impact of ART in genomic imprinting and long-term health.

## 1. Introduction

In the last century, fertility rates all over the world have experienced a significant decline. Estimations suggest that the decline will continue and the fertility rate will be below the replacement rate by 2060 [1,2]. The causes of this decline are multiple including new life style choices [3], but also an increased rate of infertility [4]. Infertility is commonly caused by lack of regular ovulation, blocked or damaged fallopian tubes, endometriosis, lack of sperm, and abnormal sperm [4]. For example, sperm counts have declined between 50 and 60% between 1973 and 2011 in North America, Europe, Australia, and New Zealand [5]. Some studies have suggested an association between some environmental stressors (like pesticides, diet, stress, smoking, pollution, and BMI) and sperm count decline, menstrual cycle alterations, and abnormal oocyte maturation [6,7,8,9,10,11,12,13,14]. In this context, assisted reproductive technologies (ART) may become the preferred option for many infertile couples. ART procedures include the conventional in-vitro fertilisation (IVF) and the intracytoplasmic sperm injection (ICSI) but also controlled ovarian hyperstimulation (COH), oocyte retrieval, embryo culture, and embryo transfer. Since its introduction in 1978, more than eight million babies have been born from IVF around the world. The number of ART cycles per year is estimated in more than two million, leading to more than 500,000 babies per year around the world [15,16]. The International Committee for ART reported in 2016 an average increase in ART cycles per year of 9.1% [16]. Since the beginning, ART procedures have been controversial and, although ART procedures are considered safe, epidemiological studies have reported an increased incidence of obstetric, perinatal, and postnatal complications in children conceived by ART [17,18,19,20]. Animal studies have also observed alterations in embryonic development, gene expression, and DNA methylation after ART [21]. In humans, epigenome studies have generally revealed no global DNA methylation alterations, but have described some regions to be especially sensitive to the stress induced by ART like imprinted regions [22]. Besides, children conceived by ART have shown higher frequency of developmental disorders caused by the disruption of genomic imprinting than children conceived naturally [20]. However, only five out of eight epigenome studies identified an enrichment of alterations at imprinted regions, which might be caused by the small sample size analysed and the sensitivity of the method used. Therefore, despite the evidence of genomic imprinting alterations after ART in humans, animal models, and farm animals, it is still unclear how ART procedures can affect genomic imprinting and to what extent. Given the critical role of imprinted genes in growth, development, and metabolism and the possible influence of these alterations during embryonic development in long-term health, it is crucial to determine the impact of ART in genomic imprinting.

This review summarises epidemiological evidence of phenotypic alterations in individuals conceived by ART and experimental evidence of the impact of the stress induced by ART in embryonic development. In addition, experimental evidence of DNA methylation alterations at imprinted regions and genome-wide level in individuals conceived by ART are discussed. Finally, current knowledge of the potential influence of genomic imprinting alterations induced by ART in postnatal growth and development and risks of long-term health conditions is summarised.

## 2. Phenotypic Outcomes Described in ART Children

The global pregnancy rate after ART was estimated to be approximately 26%. Among them, around 20% suffered early pregnancy loss [16]. From the pregnancies that reached term, epidemiological data indicate a higher risk of obstetric and perinatal complications, postnatal growth and development alterations, cancer risk and metabolic alterations in comparison with naturally conceived (Table 1).

Obstetric and perinatal complications have been widely studied in large cohorts of samples. The largest study compared 11,347 children conceived by ART with 571,914 children conceived naturally, finding significant low birth weight and preterm birth, with odd ratios (OR) of 1.13 (95%CI, 1.02–1.25) and 1.15 (95%CI, 1.06–1.25), respectively (Table 1) [23]. The largest register for ART published in 2020 also reported 1.1% extreme preterm birth (20–27 gestational weeks at delivery) and 2.2% very premature birth (28–32 gestational weeks at delivery) in ART singletons [24]. These results were also confirmed by other studies (Table 1). Large postnatal studies in children conceived by ART revealed increased risk to cerebral palsy (OR 2.22; 95% CI, 1.35–3.63) [25] and no increased risk of autism spectrum disorders [25,26,27,28,29,30]. However, a recent meta-analysis of 11 studies with large sample size found increased risk of autism spectrum disorders in children conceived by ART (RR 1.35; 95% CI, 1.09–1.68) [28]. The postnatal growth and development did not show significant differences [31,32,33,34,35,36,37], except for a significant gain of weight, height, and BMI in late infancy [38]. The risk of childhood cancer in children conceived by ART is controversial. The larger and more recent studies have found an increased risk for hepatoblastoma (SIR, 3.64; 95% CI, 1.34–7.93), rhabdomyosarcoma (SIR, 2.62; 95% CI, 1.26–4.82), central nervous system tumours (OR 1.44; 95% CI, 1.01–2.05), and malignant epithelial neoplasms (OR 2.03; 95% CI, 1.06–3.89) in children conceived by ART [39,40]. In addition to this, a meta-analysis published in 2020 including 27 studies revealed significant increased risk to paediatric cancer after frozen embryo transfer but no increased risk after other fertility treatments [41].
life-11-00728-t001_Table 1Table 1Obstetric, perinatal, and postnatal phenotypic outcomes in individuals conceived by ART.
CasesControlsNo. CasesNo. CtrlsYOB/AgeOutcomesReferencesObstetric and perinatalIVFGP33051,505,7241982–1995 Very low birth weight (<1500 g) (4.39; 95% CI, 3.62–5.32) Very preterm birth (<32 weeks) (3.54; 95% CI, 2.90–4.32) Bergh et al., 1999IVF/ICSISC2373381998–2003Preterm birth (P < 0.01)Buckett et al., 2007IVF/ICSISC74216,5251989–2006Stillbirth (4.08; 95% CI, 2.11–7.93)Wisborg et al., 2010IVF/ICSISC287648821994–2006 Low birth weight (1.4; 95% CI, 1.1–1.7) and preterm birth (1.3; 95% CI, 1.1–1.6)Henningsen et al., 2011IVFSC11,347571,9142002–2006 Low birth weight (1.13; 95% CI, 1.02–1.25) and preterm birth (1.15; 95% CI, 1.06–1.25)Sazonova et al., 2011ARTSC4333295,2201986–2002Low birth weight (P < 0.001) and preterm birth (P < 0.001)Davies et al., 2012IVF/ICSISC 181318131999–2007 Association between maternal characteristics and lower birth weightSeggers et al., 2016IVFSC177833,5552005–2014Low birth weight (RR 1.47; 95% CI 1.20–1.80) and preterm birth (RR 1.51; 95% CI 1.28–1.78)Rahu et al., 2019Postnatal growth and development IVFSC666612–45 monthsNo significant differences in developmental indicesBrandes et al., 1992IVFGP258
6–13 yearsNo significant differences in surgical procedures, malformation, height and weight, and school performanceOlivennes et al., 1997IVF + CESC158/1601560–18 monthsNo significant differences in growth features, major malformations and the prevalence of chronic diseasesWennerholm et al., 1998IVF/ICSISC9354884–6 yearsNo significant differences in motor and cognitive developmentPorjaert-Kristoffersen et al., 2005ICSISC1501478 yearsNo significant differences in pubertal staging, neurological examination, remedial therapy or surgery or hospitalizationBelva et al., 2007IVFSC-SFP1931990–4 yearsIncreased risk of lower weight, height and BMI at 3 months. Greater gain in weight (P < 0.001), height (P = 0.013) and BMI (P = 0.029) during late infancy (3 mo-1y)Ceelen et al., 2009IVF/ICSISC3091730–12 years No significant differences in head circumference, height and weightBasatemur et al., 2010ARTSC361735,848>4 yearsAn increased risk of cerebral palsy (2.30; 95% CI, 1.12–4.73)Zhu et al., 2010ARTSC33,139555,7284–13 yearsNo increased risk of autism spectrum disordersHvidtjorn et al., 2011ARTSC4333295,220<5 yearsAn increased risk of cerebral palsy (2.22; 95%CI, 1.35–3.63)Davies et al., 2012ARTSC3491847>2 yearsNo increased risk of autism spectrum disordersGrether et al., 2013ARTSC416416,5822–16 yearsNo increased risk of autism spectrum disordersLehti et al., 2013ART/OI/IISC96824710–3 yearsNo significant differences in growth, motor and cognitive developmentYeung et al., 2016IVF/ICSISC2914208,7461994–2002An increased risk of cerebral palsy (2.60; 95% CI, 1.60–4.00)Goldsmith et al., 2017ARTSC46,24910,702,377
Meta-analysis 11 studies. An increased risk of autism spectrum disorders (RR1.35; 95% CI, 1.09–1.68)Li et al., 2017Cancer riskIVFGP33051,505,7240–14 yearsNo increased risk for childhood cancerBergh et al., 1999IVFGP332
5–8 yearsNo increased risk for childhood cancerLerner-Geva et al., 2000ARTGP5249
0–15 yearsIncreased risk of childhood cancer (SIR, 1.39; 95% CI, 0.62–3.09)Bruisma et al., 2000ART SC-SFP948475321–14 yearsNo increased risk for childhood cancerKlip et al., 2001IVFGP26,692
>2 yearsIncreased risk of childhood cancer (SIR,1.42; 95% CI,1.09–1.87)Kallen et al., 2010IVFGP106,013
0–15 yearsIncreased risk of hepatoblastoma (SIR, 3.64; 95% CI, 1.34–7.93) and rhabdomyosarcoma (SIR, 2.62; 95% CI, 1.26–4.82)Williams et al., 2013ARTSC61,693351,5369–14 yearsIncreased risk for central nervous system tumours (1.44; 95% CI, 1.01–2.05) malignant epithelial neoplasms (2.03; 95% CI, 1.06–3.89)Sundh et al., 2014IVF/ICSI/FETSC2549/81,450/25,5631,393,284
Meta-analysis 27 studies. Increased risk to childhood cancer after FET (1.37; 95% C, 1.04–1.81). Zhang et al., 2020Metabolic effectsIVF/ICSI + CEGP69714–10 years↑HDL, ↓triglycerides, ↑IGF-2, ↑height. Normal body fat and fasting glucoseMiles et al., 2007IVFSC-SFP2332338–18 years↑ Body fat, ↑ blood pressure, ↑ fasting glucose Ceelen et al., 2007 & 2008IVFSC106684–14 years↑Blood pressure, ↑triglycerides, ↑TSH. Normal fasting glucoseSakka et al., 2009 & 2010ICSIGP21722314 years↑ Body fat normal blood pressure Belva et al., 2012bARTSC65577–18 yearsSystemic and pulmonary vascular dysfunction Scherrer et al., 2012IVFSC10102–4 weeks Subclinical hypothyroidism Onal et al., 2012ARTSC50506 monthsCardiac and vascular remodelling at both time pointsValenzuela-Alcaraz et al., 2013IVFSC-SFP63794 years↑ Blood pressure, ↑body fat La Bastide-Van Gemert et al., 2013IVFSC-FP142017–26 years ↓Peripheral insulin sensitivity Chen et al., 2014IVF/ICSISC-FP282205–6 years↑Fasting glucose Pontesilli et al., 2015ARTSC54547–18 yearsRight ventricular dysfunction von Arx et al., 2015IVF/ICSISC211240961 month–12 yearsMeta-analysis 19 studies. ↑ blood pressure ↓ LDL ↑fasting insulin levels. Suboptimal cardiac diastolic function and ↑ vessel thickness. Guo et al., 2017ART, assisted reproductive technologies; IVF, in vitro fertilisation; ICSI, intracytoplasmic sperm injection; OI, ovulation induction; II, intrauterine insemination; SC, spontaneous conception; GP, general population; SFP, subfertile parent; FP, fertile parent; CE, cryopreserved embryos; CI, confidence intervals; RR, prevalence proportion ratios; IGF, insulin-like growth factor; BMI, body mass index; HR, hazard ratio; SIR, standardized incidence ratio; BMD, bone mineral density; BMI, body mass index; CVD, cardiovascular disease; HDL, high-density lipoprotein cholesterol; LDL, low density lipoprotein cholesterol; TG, triglyceride; SGA, small for gestational age; TSH, thyroid stimulating hormone. Furthermore, children and adolescents conceived by ART showed increased risk of metabolic and vascular alterations such as higher body fat, blood pressure, hypothyroidism, and vascular dysfunction (Table 1) [42,43,44,45,46,47,48,49,50,51,52,53]. A meta-analysis of 19 studies reported an increased risk of higher blood pressure and fasting insulin levels, lower LDL, and suboptimal cardiac diastolic function with higher vessel thickness after ART [54]. The presence of these metabolic alterations in childhood and adolescence may predispose them to chronic diseases like obesity, type 2 diabetes, and cardiovascular disorder in adulthood [55,56].

## 3. ART Can Induce Stress in the Embryo

The complications described in ART children are probably caused by the significant differences between in vitro and in vivo environments. The development of the embryo in the female reproductive tract involves the exposure to multiple hormones, nutrients, growth factors, and cytokines and well-maintained environmental conditions [45]. In contrast, ART procedures require the extraction and in vitro culture of gametes and embryos, the manipulation through pipetting, and the exposure to temperature, oxygen, and pH oscillations [57]. Besides, these stressors can occur individually or simultaneously [58] and the stress induced in the embryo can affect developmental rates and embryo quality [21].

ART procedures take place during gametogenesis, fertilisation, and early embryo development including pre-implantation and implantation stages (Figure 1). 

Therefore, ovarian hyperstimulation (COH) and gamete freezing occur during gametogenesis, IVF during fertilisation, embryo culture and freezing during preimplantation and preimplantation genetic diagnosis (PGD), and assisted hatching and embryo transfer during the implantation stage [59]. The ovarian hyperstimulation or superovulation is a common procedure used to obtain a large number of oocytes to increase ART success rates. Generally, superovulation produces oocytes morphologically and functionally preserved [60,61], however, adverse effects like alterations in oocyte maturation rate have been reported [62,63]. Superovulation in mice can produce a delay in embryonic and foetal development and decreases implantation rates [64,65]. Likewise, superovulation was associated with alterations in the expression of some genes in the oocyte, producing a lower quality of oocytes and embryos in mice [66]. Superovulation is usually followed by in vitro fertilisation (IVF) or intracytoplasmic sperm injection (ICSI). The stress induced specifically by these is difficult to estimate because in a normal situation, these are preceded by superovulation and followed by in vitro culture. Nevertheless, mice studies described slightly delayed in prenatal and postnatal development with lower birth weight and smaller placentas than those naturally conceived [53,67]. Later in development, IVF adult mice showed impaired glucose tolerance, increased fasting glucose levels, and reduced insulin-stimulated Akt phosphorylation in the liver [53]. These results are similar to the ones obtained in humans, suggesting that these alterations may be caused by ART procedures. ICSI can also have additional complications. In this procedure, sperm is directly injected into the egg, thus avoiding the natural selection of the sperm based on motility, acrosome activation, and membrane fusion of gametes. The alterations in the sperm can lead to additional alterations in the embryonic development [68,69] and might explain why ICSI is the fertilisation procedure that induced the highest number of differentially expressed genes [22].

After in vitro fertilisation, the fertilized egg is cultured in vitro. This procedure has been widely used in animal models, farm animals, and humans, however, culture conditions are still not completely optimised. The two main factors that can affect embryo development during in vitro culture (IVC) are the culture conditions and the culture medium composition. Therefore, mouse blastocysts after in vitro culture in different culture conditions show significant differences in expression levels, with an enrichment in genes associated with metabolism [70]. Among the culture media tested, Whitten’s medium at 20% oxygen is the most stressful culture condition inducing a high number of differentially expressed genes [22]. Apart from gene expression changes, some studies have also reported epigenetic changes after IVC, possibly caused by alterations in epigenetic reprogramming during the transition from the zygote to blastocyst stage (Figure 1) [21,71,72,73,74].

## 4. ART, Epigenetic Reprogramming, and Genomic Imprinting

During embryonic development, DNA methylation plays a crucial role in cell differentiation, sex chromosome dosage compensation, retrotransposons repression, and genomic imprinting, but it is also essential for the success of sexual reproduction in mammals [75]. During gametogenesis, primordial germ cells erase somatic DNA methylation signatures and establish sex-specific and germ cell-specific epigenetic signatures (Figure 1). This first wave of epigenetic reprogramming is required to establish the appropriate imprinting marks in germ cells [76]. Maternal marks are established in oocyte and paternal marks in sperm (Figure 1). After fertilisation, the zygote requires the erasure of DNA methylation marks to initiate the embryonic development. In this second wave of epigenetic reprogramming, genomic imprinting, transposable elements, and metastable epialleles should be protected from demethylation (Figure 1) [76]. ART procedures occur simultaneously to this extensive epigenetic reprogramming and the stress produced by ART might affect the establishment and/or maintenance of genomic imprinting.

Genomic imprinting is an epigenetic process that is known to affect the expression of more than 100 human genes with crucial roles in growth, metabolism, and development including the control of embryonic growth, placenta development, and post-natal development and metabolism [77,78,79,80]. Genomic regions controlled by genomic imprinting present differential imprinting marks to distinguish maternal and paternal alleles. These marks cause monoallelic expression in a parent-of-origin-specific manner. The differential epigenetic marks that control allele-specific expression are established during gametogenesis in male and female germline (Figure 2). Imprinted genes are usually organised in clusters called imprinted domains, which share regulatory elements such as differentially methylated regions (DMRs) and imprinting control region (ICR) (Figure 2). The disruption of a single or multiple germline DMRs (gDMRs) affecting the expression of imprinted genes can cause congenital imprinting disorders (CIDs). CIDs are a group of developmental disorders with overlapping clinical features such as growth abnormalities, metabolic alterations, and developmental delay [81]. Genomic imprinting alterations have also been implicated in complex disorders such as autism and non-syndromic embryonal tumours (e.g., Wilms tumour) [82,83].

Errors in the establishment and/or maintenance of the imprinting have been found in hydatidiform mole [85,86,87], miscarriages, congenital imprinting disorders (CIDs), and multi-locus imprinting disturbances (MLIDs) [88,89,90]. Women with mutations in *NLRP7* showed hydatidiform moles and miscarriages, caused by loss of methylation at multiple ICRs [86]. Likewise, maternal mutations in components of subcortical maternal complex (SCMC) such as NLRP5 and NLRP2 have been described in CID cases with MLIDs [91]. SCMC components are expressed in mammalian oocytes and early embryos before zygote genome activation and might be responsible for the availability of the components required for the protection of genomic imprinting during epigenetic reprogramming [92]. One important component of this protection is ZFP57, a zygotic factor in humans, which recognizes methylated CpG in the TGCC^me^GC hexamer consensus motif located in some ICRs. ZFP57 recruits TRIM28 and KAP1 complexes, which promote the recruitment of SETDB1 and DNMT1 to maintain the methylation at those ICRs [93]. Recessive mutations in *ZFP57* have been identified in transient neonatal diabetes cases, a congenital imprinting disorder, with a specific pattern of MLIDs [94,95].

Consequently, the disruption of the establishment and maintenance of imprinting marks by the stress induced by ART might affect normal embryonic development and future health.

## 5. Genomic Imprinting Alterations after ART

Numerous studies have reported an increased risk of congenital imprinting disorders in children conceived by ART [20,96,97]. In 2002, Cox et al. reported two Angelman syndrome (AS) cases conceived by ICSI with both loss-of-methylation (LOM) at maternal SNURF:TSS DMR, one with complete LOM and the other one with partial LOM [98]. Interestingly, the frequency of this epimutation in AS conceived naturally was less than 5% of cases, suggesting a possible link between ICSI and AS. In 2003, three different studies in three different countries, the UK, the U.S., and France, observed higher frequency of BWS cases after ART finding ~4, ~6, and ~3.2-fold increase, respectively [19,99,100]. The majority (>90%) of BWS cases conceived by ART showed LOM at KCNQ1OT1:TSS DMR, in comparison with 50% of BWS cases with this epimutation in the naturally conceived. In 2006, a preliminary British survey with 213 BWS, 38 TNDM, 384 AS, and 522 PWS observed significant increased frequency of BWS in children conceived by ART [101]. In 2014, a nationwide epidemiological study of the Japanese population revealed that the frequencies of imprinting disorders after ART was 1.6% for AS, 1.5% for PWS, 8.6% for BWS, and 9.5% for SRS, respectively [97]. The results showed a slight increment in the frequency of AS and PWS patients after ART, but 10-fold to 12-fold greater than expected in BWS and SRS patients [97]. Finally, in 2019, another nationwide epidemiological study in Japan found 3.44-, 4.46-, and 8.91-fold increased frequencies of PWS, BWS, and SRS after ART, respectively [20]. Similarly, in livestock, LOM at IGF2R imprinting DMR has been described after embryo culture, causing a rare overgrowth syndrome called large offspring syndrome (LOS) with similar phenotypic features than BWS including overgrowth, macroglossia, umbilical hernia, and visceromegalia [102].

The comparison of clinical phenotype and molecular features of ART (IVF and ICSI) and non-ART children with sporadic BWS revealed lower frequency of exomphalos and a higher risk of neoplasia [103]. This study also observed LOM at other imprinting control regions (ICRs) in 37.5% of ART and 6.4% of non-ART BWS IC2 cases. Likewise, the study of DNA methylation changes at 23 gDMRs in CID-ART cases revealed a mixture of mild hypermethylation and hypomethylation in maternal and paternal gDMRs [97]. These observations suggested MLIDs with cellular mosaicism, which might indicate errors in the imprinting after fertilisation, probably by the impairment of imprinting maintenance [97]. Likewise, another study performed in 2019 also observed incomplete and more widespread DNA methylation variations in SRS cases conceived by ART than those conceived naturally [20]. In this study, aberrant DNA methylation patterns appeared at multiple imprinted regions in both maternal and paternal gDMRs, with both hyper and hypomethylation events and in mosaic [20]. How ART can affect the protection of gDMRs during epigenetic reprogramming is still unknown, however, mice studies revealed that DNA methylation at imprinted regions might be particularly sensitive to culture medium conditions [71,104,105]. For example, the exposure to 5% or 20% oxygen during in vitro culture of IVF embryos produced abnormal DNA methylation and expression of imprinted genes in the placentas [106]. In addition, mice embryos cultured in Whitten’s medium showed biallelic expression of the *H19* gene caused by loss of DNA methylation on the paternal allele [104] and embryos cultured in vitro in different commercial culture media showed loss of imprinting at H19, Snrpn, and Peg3 *DMRs* [107]. All these observations indicate that the incidence of imprinting alterations in humans, animal models, and farm animals after ART might be higher than expected by chance.

## 6. Epigenome Studies in Children Conceived by ART

The stress induced in the embryo by ART might affect DNA methylation at imprinted regions, but the impact in the epigenome is still unclear. Table 2 summarises the epigenome studies performed with methylation array platforms. The comparison of umbilical cord blood and placenta DNA methylation profiles of 10 ART cases and eight naturally conceived (NC) revealed 733 differentially methylated probes (DMPs) [108]. This study also detected.

From these studies, we can conclude that no global DNA methylation changes were observed in children conceived by ART in comparison with children conceived naturally. DNA methylation changes detected were mild and widespread. The enrichment of DNA methylation alterations at imprinted regions was detected in six of eight studies performed, however, each study found different iDMRs affected with the exception of the *GNAS* locus. The discrepancies found between studies can be explained by the low sample size analysed, sample type, confounding factors used for correction, and threshold of significance used. In addition, the array platform only partially covered some genomic regions, in particular imprinted regions (for example, Illumina^®^ Infinium Human Methylation450 array covered IGF2:alt TSS DMR and MEG3/DLK1:IG DMR with only a single probe) and this will affect the detection of methylation alterations. Additionally, the efficiency of the methylation array platform to detect mild DNA methylation changes can be limited, as we observed when comparing the Illumina^®^ Infinium Human Methylation450 array and a hybridisation custom bisulphite sequencing panel of imprinted regions [115]. However, not all epigenome studies detected an enrichment of imprinting alterations; most of these studies identified an apparent enrichment and pyrosequencing studies detected significant imprinting alterations in children conceived by ART (cord blood and peripheral blood) [116,117]. The apparent alteration of DNA methylation at imprinted regions [108,109,110,112] might explain why the frequency of congenital imprinting disorders in children conceived by ART is greater than expected in the general population [20]. Besides, the imprinting alterations reported were mild, suggesting some level of mosaicism, which might indicate that changes occur after fertilisation, during the transition from zygote to blastocyst, when genomic imprinting should be protected during epigenetic reprogramming.

Nevertheless, only a percentage of children conceived by ART showed phenotypic outcomes and mild and widespread DNA methylation alterations [20]. In this context, the identification of DNA methylation changes induced by ART could be problematic with small sample size, methylation assays with low sensitivity and low number of probes per region, and bioinformatic pipelines designed for other purposes. In this sense, the approximation proposed by Choufani et al. (2019) with the identification of the ART “outlier” group is promising [112]. The ART outlier group showed 84,270 DMPs with an enrichment of alterations in imprinted regions in comparison with those naturally conceived. In contrast, the comparison of the whole ART group with those naturally conceived showed no significant alterations. Therefore, understanding the impact of ART at the DNA methylation level will require the largest set of samples per subgroup (IVF/ICSI, fresh/frozen) with similar culture conditions and adapted bioinformatic pipelines that allow for the identification and study of this “outlier” ART group [112]. Studying the effect of ART in genomic imprinting will require more sensitive approaches capable of interrogating more CpGs per region and with high coverage. In this sense, two new methods have been developed in the last years, Implicon [118] and ImprintSeq [115], which are capable of interrogating 14 and 63 iDMRs, respectively, and with more than 100X of coverage.

## 7. Implications in Long-Term Health

The comparison of epigenome changes induced by ART at birth and adulthood showed that changes at birth mostly resolved in adulthood [114]. Nevertheless, the removal of these changes in adulthood may not avoid the consequences of the adaptation induced in the embryo to those changes that occurred during embryonic development. Embryos have the ability to adapt and adjust to variations in their environment through developmental plasticity [119]. The adaptations and alterations that occur during embryo and foetal development may induce wide range of conditions and phenotypes in adulthood, as the Developmental Origins of Health and Disease (DOHaD) hypothesis proposed [120,121]. Therefore, the environmental perturbations induced by ART in the embryo that can affect developmental speed, metabolism, gene expression, and DNA methylation might influence the long-term health. However, it is still unknown if these alterations can translate into adaptation responses that lead to long-term consequences (Figure 3).

Some observations suggest that imprinted regions might be more susceptible than other genomic regions to be affected by ART. The alteration of genomic imprinting during embryonic development can affect embryo implantation and embryo development [71] but the effect of these alterations in long-term health is still unclear. Mice studies have revealed alterations in body composition, lipid metabolism, blood pressure, and glucose tolerance by the restriction of imprinted genes at prenatal and early postnatal stages [122]. Likewise, the in vitro culture of mouse embryos has been associated with higher risk to obesity, anxiety, and memory deficits [71,123]. In humans, individuals with congenital imprinting disorders like BWS and SRS may also suffer some long-term health conditions. For example, Silver–Russell Syndrome presents growth failure, severe feeding difficulties, gastrointestinal problems, hypoglycaemia, body asymmetry, and scoliosis as well as motor and speech delay [124]. In addition, SRS cases can describe increased risk of metabolic disorders after rapid postnatal weight gain, premature adrenarche, early and rapid central puberty, and insulin resistance [124]. In contrast, Beckwith–Wiedemann Syndrome shows overgrowth, macroglossia, exomphalos, umbilical hernia, and childhood cancer [125]. In adulthood, individuals with BWS show increased risk of neoplasia, infertility, and renal anomalies [126]. Meanwhile, some individuals conceived by ART have shown low birth weight, higher weight, height, and BMI in late infancy [38] and higher blood pressure and fasting insulin levels, lower LDL, and vascular dysfunction in adolescence [54]. This development delay at birth, probably caused by epigenetic alterations enriched in imprinted regions, followed by accelerated growth in late infancy and metabolic alterations in adolescence, may predispose them to chronic diseases like obesity, type 2 diabetes, and cardiovascular disorder in adulthood [127,128].

Metabolic imprinting term was adopted by Waterland and Garza in 1999 [129] and refers to the relationship between imprinting alterations during early embryonic development and consequences later on in the development [130,131]. This hypothesis proposed the epigenetic programming of metabolism during prenatal and postnatal periods as a response to imprinting alterations that occurred during early development. For example, children from pregnant women exposed to famine during the Dutch Hunger Winter showed loss of DNA methylation at the *IGF2* imprinted gene locus and an increased risk of diabetes, obesity, cardiovascular disease, and other health problems [132]. In this case, pregnant women exposed to famine showed imprinting alterations in the embryo that affected long-term health later on in the development. In the case of ART, the developmental restriction during early or mid-gestation might be followed by accelerated placental and foetal growth, which might lead to cardiometabolic alterations during adulthood [133]. Adolescents conceived by ART showed significant differences in growth kinetics, glucose levels, fat deposition, and blood pressure in comparison with those conceived naturally by subfertile parents [38,51,52]. The impact of ART in adulthood is still unknown, because the first IVF baby was born in 1978, however, all of these observations suggest that early-life environmental conditions induced by ART, which might promote epigenetic changes and imprinting alterations, may predispose long-term conditions (Figure 3).

## 8. Conclusions

ART procedures can induce stress in the embryo and affect the epigenome, but not globally. Some authors have described mild widespread DNA methylation changes at imprinted regions, which might suggest that the alterations occur after fertilisation by the disruption of the maintenance of imprinting during epigenetic reprogramming. These changes in imprinted regions might also induce epigenetic programming of metabolism during the prenatal and postnatal periods with significant consequences in long-term health. However, uncovering the real impact of ART on human health and the implication of the genomic imprinting on it will require further and larger studies. Given the growing use of ART, it is important to determine its impact in long-term health to improve procedures and reduce risks.

## Figures and Tables

**Figure 1 life-11-00728-f001:**
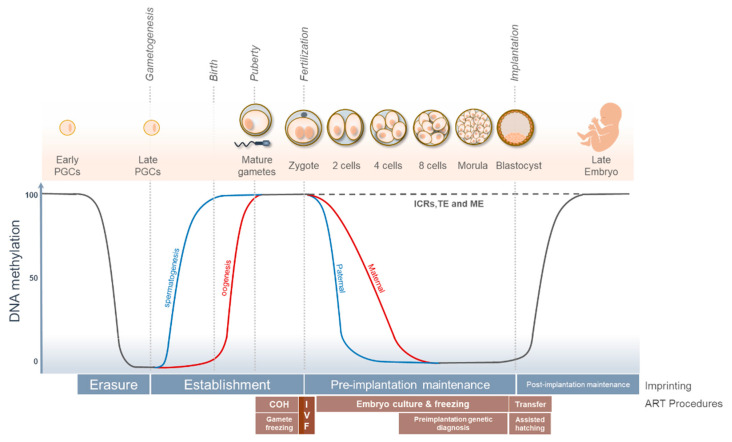
Life cycle of genomic imprinting in humans, epigenetic reprogramming, and ART procedures. PGCs, primordial germ cells; COH, controlled ovarian hyperstimulation; IVF, in vitro fertilisation; ART, assisted reproductive technology; ICRs, imprinting control regions; TE, transposable elements; ME, metastable epialleles.

**Figure 2 life-11-00728-f002:**
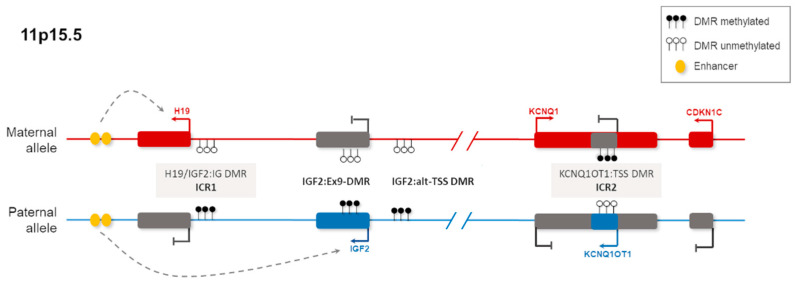
11p15.5 imprinted region contains four iDMRs, two germline DMRs (H19/IGF2:IG DMR paternally imprinted and KCNQ1OT1:TSS DMR maternally imprinted), and two secondary DMRs (IGF2:Ex9 DMR and IGF2:alt-TSS DMR). In this case, both gDMRs are also considered as imprinting control regions (ICRs). This region is associated with two congenital imprinting disorders, BWS and SRS. The loss-of-methylation of ICR2 or the gain-of-methylation of ICR1 causes BWS, an overgrowth disorder. In contrast, the loss-of-methylation of ICR1 causes SRS, a growth retardation disorder. DMR, differentially methylated region; ICR, imprinting control region [84].

**Figure 3 life-11-00728-f003:**
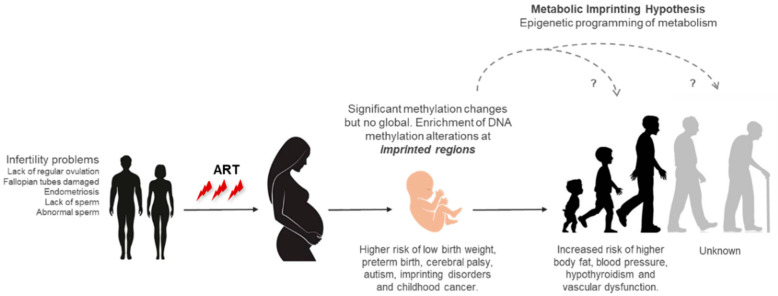
Diagram of the possible impact of ART in long-term health.

**Table 2 life-11-00728-t002:** Epigenome studies performed with methylation array platforms in individuals conceived by assisted reproductive technologies.

Cases	Controls	No. Cases	No. Ctrls	Sample Type	Platform	No. DMPs ^1^	iDMRs Affected	Results	Reference
IVF	SC	10	13	cord blood/placenta	Illumina GoldenGate^®^ Assay ^2^	82	GNAS, NNAT, PEG3, IGF2AS/MEST, GRB10, PEG3	ART-Placenta showed lower mean methylation and ART-cord blood higher. DNA methylation differences associated with gene expression differences at both imprinted and non-imprinted genes.	Katari et al., 2009
ART	SC	10	8	cord blood/placenta	Illumina^®^ Infinium Human Methylation27 array ^3^	733	GNAS, PLAGL1, DIRAS3, ZIM2	Significant differences in DNA methylation were enriched in certain types of genomic locations and with greater variability and more hypomethylation.	Melamed et al., 2015
ART	SC	94	43	neonatal blood spot	Illumina^®^ Infinium Human Methylation450 array ^4^	n.s.	n.s.	Significant differences in DNA methylation associated with IVF/ICSI culture conditions and/or parental infertility were detected at metastable epialleles. Imprinted genes are differentially methylated more often than expected by chance. No differences between ICSI-frozen and intrauterine insemination.	Estill et al., 2016
IVF/ICSI	SC	34/89	53	cord blood	Illumina^®^ Infinium Human Methylation450 array ^4^	4730	NAP1L5, L3MBTL, GNAS, PEG10	Significant differences in DNA methylation but with small (β < 10%) or very small (β < 1%) effect size. ICSI showed a significantly decreased DNA methylation age at birth. DMPs enriched in CpG islands with low methylation values and in ICRs.	El Hajj et al., 2017
ART	SC	23	41	cord blood	Illumina^®^ Infinium Human Methylation450 array ^4^	0 ^$^		No significant difference in DNA methylation in ART. Significant differences found between stochastic epigenetic variability and four multiple factor analysis dimensions summarizing common phenotypic, behavioral or environmental factors.	Gentilini et al., 2018
ART	SC	44	44	Placenta	Illumina^®^ Infinium Human Methylation450 array ^4^	0 ^¥^/84,270 ^¥¥^	GNAS, SGCE, KCNQT1OT1, BLCAP/NNAT ^¥¥^	No significant difference in DNA methylation comparing ART with controls. The comparison of ART outlier group with controls showed significant differences in DNA methylation, enriched in loss of methylation of several imprinted genes. IVF/ICSI showed distinct epigenetic profiles.	Choufani et al., 2019
ART	SC	193	86	neonatal blood spot/ adult blood	Illumina^®^ Infinium Human MethylationEPIC array ^5^	2340	n.s.	Significant difference in DNA methylation around birth. No difference found with embryo culture. Epigenetic variation at birth mostly resolves by adulthood. No significant association with imprinting regions but using relaxed threshold, 4% of imprinting-related probes showed differential methylation at birth.	Novakovic et al., 2019
IVF/ICSI	SC	87	70	cord blood	Illumina^®^ Infinium Human Methylation450 array ^4^	19		No significant association of ART with global methylation levels, imprinted loci and meta-stable epialleles. No difference was found between IVF and ICSI. DMPs map to genes related to brain function/development or genes connected to conditions linked to subfertility.	Tobi et al., 2021

^1^ FDR < 0.05; ^2^ This platform interrogated 1536 CpGs; ^3^ This platform interrogated 27,578 CpGs; ^4^ This platform interrogated 485,000 CpGs; ^5^ This platform interrogated 850,000 CpGs; ^¥^ ART vs controls; ^¥¥^ ART outlier group vs. controls; ^$^ significance threshold genome-wide approach (*p* < 10^−7^); DMP, differentially methylated probes; iDMR, imprinting differentially methylated regions; IVF, in vitro fertilisation; ICSI, intracytoplasmic sperm injection; ART, assisted reproductive technology; ICR, imprinting control regions; β, beta-value; *n*.s. not specified.an enrichment in certain genomic locations, in particular imprinted regions and observed high variability [108]. The study of neonatal blood spots from 94 ART and 43 NC showed no global methylation alterations and an enrichment of alterations in imprinted genes and metastable epialleles [109]. The comparison of cord blood from 123 ART and 53 NC identified 4730 DMPs, but with small (β < 10%) or very small (β < 1%) effect size [110]. This study also reported an enrichment in CpG islands with low methylation values (0–20%) and located in imprinting control regions (ICRs) [110]. Interestingly, this study observed a significantly decreased DNA methylation age at birth, approximately half a week behind, in the ICSI group. In contrast, a similar approach with 23 ART and 41 NC found no significant differences in children conceived by ART and children conceived naturally [111]. However, this work applied a more stringent significance threshold, genome-wide approach (*p* < 10^−7^) instead of *p*-value adjusted by FDR < 0.05. The authors of this work suggested that the number of stochastic epigenetic variations induced by ART was not greater than the ones induced naturally in response to maternal behaviour or other common environmental factors [111]. Furthermore, the study of 44 placentas from both children conceived by ART and children conceived naturally found no difference in DNA methylation after ART [112]. However, using principal component (PCA) of the whole placenta dataset (414,320 CpGs), they determined an “outliers” group with 11 ART and four controls. This ART outlier group showed significant differences in DNA methylation and an enrichment in loss of methylation of several imprinted genes [112]. Another study of cord blood from 87 ART and 70 NC found no global methylation changes after ART and identified 19 DMPs in the ART group [113]. This study found no difference between IVF and ICSI or association with imprinted loci and meta-stable epialleles. Finally, the largest study until now explored the impact of epigenetic changes induced by ART at birth and adulthood using a cohort of 193 ART and 86 NC (neonatal blood spots and adult blood from the same individual) [114]. This study identified 2340 DMPs at birth that mostly resolve by adulthood [114]. No significant association was detected in imprinting regions, however, using a relaxed threshold, 4% of imprinting-associated probes showed differential methylation in neonatal blood.

## Data Availability

Not applicable.

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
