# Peer review of "Alteration of Genomic Imprinting after Assisted Reproductive Technologies and Long-Term Health"

_life, 2021, doi:10.3390/life11080728_

Round 1
Reviewer 1 Report
Presented manuscript describes the extent and the effects of epigenetic changes in the human genome resulting from the application of assisted reproductive technologies (ART) in cases of infertility in humans. Manuscript is well written and tackles interesting topics relevant to molecular background of undesired side-effects of ART solutions. It clarifies the importance of imprinting regions of the human genome being the main concern of further epigenetic research on fertility disorders. Presented content might meet with larger interest including scientific and ethical background.
Author Response
Presented manuscript describes the extent and the effects of epigenetic changes in the human genome resulting from the application of assisted reproductive technologies (ART) in cases of infertility in humans. Manuscript is well written and tackles interesting topics relevant to molecular background of undesired side-effects of ART solutions. It clarifies the importance of imprinting regions of the human genome being the main concern of further epigenetic research on fertility disorders. Presented content might meet with larger interest including scientific and ethical background.
RESPONSE: Thank you for your comments.
Reviewer 2 Report
Ochoa has submitted a review on the relevance of genomic imprinting for assisted reproduction technologies (ART) and long-term health.
In principle, this is an interesting topic, but I´ve two major concerns about the current version:
- a) the author scarcely provide information on long-term health in individuals born after ART. The author should either change the title or add further findings (see b)).
- b) the author should include further studies on the frequencies of ART-born children among imprinting disorders cohorts (e.g. from the E. Maher group). The author only briefly mentions frequencies among some of these entities, but for the reader it would be valuable to see more data to decide whether disturbed imprinting with clinical relevance has to be considered in the context of ART or not. Accordingly, in case of disorders like BWS and SRS, the authors might transfer the already known long-term health issues of these disorders in his discussions on long-term health issues.
Further comments:
- Abstract: the authors mentions imprinting disorders in general, but a more precise information should be given: to the best of my knowledge, an effect of ART has only been suggested for BWS, SRS and PWS?
- Introduction: environmental stress: the author only refers to male environmental stressors, but should also briefly include female factors
- 2. Phenotypic outcome in ART: second line: it is important to present the ration of pregnancy loss among ART pregnancies, but what is the rate among non-ART pregnancies?
- page 2, same paragraph: increased risk for hepatoblastoma: a finding suggestive for BWS, thus are their data that the ART-conceived children with hepatoblastoma suffer from BWS?
- 4. ART, epigenetic reprogramming:
Second paragraph: the author generally describes that “paternally imprinted are only maternally expressed” and vice versa. In fact, this is a general observation, but exceptions exist, and the author shows one in figure 2 (IGF2). Thus, this sentence should be modified.
- page 6, ZFP57: “a critical component of protection” is a very general and should be more precise: are all gDMRs protected, what does protection mean?
- Conclusion: too long, and includes several further aspects not covered by previous paragraphs. Thus it should either be shortened, or renamed. Conclusion is only the last paragraph.
- figures: the appear to be leaned on already published figures, should this be indicated in the legend?
Author Response
Ochoa has submitted a review on the relevance of genomic imprinting for assisted reproduction technologies (ART) and long-term health. In principle, this is an interesting topic, but I´ve two major concerns about the current version:
- the author scarcely provide information on long-term health in individuals born after ART. The author should either change the title or add further findings (see b)).
RESPONSE: Thank you for your comment. Further findings were added to new version of the manuscript. Please see section 7: Implications in long-term health.
- the author should include further studies on the frequencies of ART-born children among imprinting disorders cohorts (e.g. from the E. Maher group). The author only briefly mentions frequencies among some of these entities, but for the reader it would be valuable to see more data to decide whether disturbed imprinting with clinical relevance has to be considered in the context of ART or not. Accordingly, in case of disorders like BWS and SRS, the authors might transfer the already known long-term health issues of these disorders in his discussions on long-term health issues.
RESPONSE: Thank you for your observation. More information was included in section 5: Genomic imprinting alterations after ART, including first descriptions of AS and BWS cases after ART and the phenotypic and molecular diagnostic of ART and non-ART BWS cases. In addition to this, long-term health conditions associated with SRS and BWS were also included in section 7: Implications in long-term health.
- Abstract: the authors mentions imprinting disorders in general, but a more precise information should be given: to the best of my knowledge, an effect of ART has only been suggested for BWS, SRS and PWS?
RESPONSE: Thank you for your observation. The first sentence of the abstract was changed to clarify this point.
- Introduction: environmental stress: the author only refers to male environmental stressors, but should also briefly include female factors
RESPONSE: Thank you for your comment. Female factors were added to the introduction.
- Phenotypic outcome in ART: second line: it is important to present the ration of pregnancy loss among ART pregnancies, but what is the rate among non-ART pregnancies?
RESPONSE: Thank you for your question. Strumpf et al. 2021 (Prevalence and clinical, social, and health care predictors of miscarriage, BMC Pregnancy and Childbirth, 21) performed a study in Manitoba population with 79,978 women finding 11.3% of miscarriages. However, the American Society for Reproductive Medicine in 2012 considered the frequency of pregnancy losses in 15–25% of pregnancies. Half of them attributed to chromosomal abnormalities. Discarding chromosomal abnormalities, to be comparable with ART pregnancies (these alterations are generally detected in preimplantation genetic diagnosis), the frequency of pregnancy loss in non-ART pregnancies is 7.5-12.5%.
- page 2, same paragraph: increased risk for hepatoblastoma: a finding suggestive for BWS, thus are their data that the ART-conceived children with hepatoblastoma suffer from BWS?
RESPONSE: Thank you for your question. This is a really interesting point. BWS is a rare overgrowth disorder with a frequency in general population of 1:15,000 births. Large epidemiological studies like the ones performed by Hiura et al. 2014 and Hattori et al. 2019 revealed a higher frequency of BWS, SRS and PWS after ART in comparison with naturally conceived. BWS is caused by the loss of imprinting at 11p15.5 and is classified based on the type of alteration: loss-of-methylation (LOM) at imprinting control region (ICR) 2, uniparental disomy (UPD) of chromosome 11, gain-of-methylation (GOM) at ICR1 and not alteration was detected (NAD). The frequency of each BWS type is the following, 50% ICR2 LOM, 20% UPD11, 7% GOM ICR1 and 17% NAD. The risk of hepatoblastoma is 0.7%, 3.5%, 0% and 0.3% respectively in each BWS type. After ART, more than 90% of BWS cases are BWS-ICR2 LOM and BWS-NAD. Carli et al. 2019 (JAMA Pediatr. 2019;173(10):996. doi:10.1001/jamapediatrics.2019.2355) estimated the magnitude of the association between ART and BWS with hepatoblastoma predisposition, finding no significant association and proposing that both embryonal tumors and BWS are a consequence of the complex epigenetic alterations in children conceived by ART. Carli et al. used the data published by Spector et al. in 2019 (JAMA Pediatr. 2019;173(6):e190392. doi:10.1001/jamapediatrics.2019.0392) with a cohort of 275,686 IVF and 2,266,847 non-IVF individuals finding hepatoblastoma in 21 and 58 individuals respectively. If the frequency of BWS is 4.46-fold increased after ART (Hattori et al. 2019), in this IVF cohort we should expect approximately 81 BWS cases. If the majority of BWS are ICR2 LOM and NAD, the expected frequency for hepatoblastoma will be 0.7% and 0.3%. However, 21 cases of hepablastoma were described in this IVF cohort, which represents 28% of the expected number of BWS cases. Lim et al. 2009 observed higher frequency of cancer in BWS-ART than in BWS conceived naturally but not as high as 28%. Unfortunately, without methylation analysis of these individuals is not possible to know if hepatoblastoma cases had alteration at 11p15.5 associated with BWS. Another consideration is that due to mosaicism the alterations might be present only in some cell types. Interestingly, Coorens et al. Science 2019 identified H19 hypermethylation (ICR1 GOM) in the kidney as an embryonal precursor of Wilms tumor in absence of BWS. The level of mosaicism and other imprinting alterations (MLIDs) should be also considered, however further analysis will be necessary to clarify this interesting discussion.
- ART, epigenetic reprogramming: Second paragraph: the author generally describes that “paternally imprinted are only maternally expressed” and vice versa. In fact, this is a general observation, but exceptions exist, and the author shows one in figure 2 (IGF2). Thus, this sentence should be modified.
RESPONSE: Thank you for your observation. The sentence was modified.
- page 6, ZFP57: “a critical component of protection” is a very general and should be more precise: are all gDMRs protected, what does protection mean?
RESPONSE: Thank you for your comment. More information of the role of ZFP57 in the protection of imprinting during epigenetic reprograming was included in the new version of the manuscript.
- Conclusion: too long, and includes several further aspects not covered by previous paragraphs. Thus it should either be shortened, or renamed. Conclusion is only the last paragraph.
RESPONSE: Thank you for your comment. In the new version of the manuscript, only the last paragraph is included in Section: Conclusions. The rest was reorganised in other sections or discarded.
- figures: the appear to be leaned on already published figures, should this be indicated in the legend?
RESPONSE: Thank you for your observation. Yes, Figure1 is based on other figures previously published [1-4] but despite the similarity of all these figures, this was not indicated in the legend of previous publications. Figure 2 was based on [3], so I added to the legend this information. Figure 3 was not based on previous publications.
- Smallwood, S. A., Kelsey, G. De novo DNA methylation: a germ cell perspective,Trends in Genetics 28, 33-42 (2012) https://doi.org/10.1016/j.tig.2011.09.004.
- Monk, D. Germline-derived DNA methylation and early embryo epigenetic reprogramming: The selected survival of imprints. The International Journal of Biochemistry & Cell Biology 67, 128-138 (2015) https://doi.org/10.1016/j.biocel.2015.04.014.
- Monk, D., Mackay, D.J.G., Eggermann, T. et al.Genomic imprinting disorders: lessons on how genome, epigenome and environment interact. Nat Rev Genet 20, 235–248 (2019). https://doi.org/10.1038/s41576-018-0092-0
- Ishida, M., Moore, G. E. The role of imprinted genes in humans. Molecular Aspects of Medicine 34, 826-840 (2013) https://doi.org/10.1016/j.mam.2012.06.009.